# Dietary diversity insufficiently explains differences in prevalence of anaemia in pregnancy across regions in Nigeria: A secondary analysis of Demographic and Health Survey 2018

Ochuwa Adiketu Babah[1,2,3☯]*, Diana Sagastume[3☯], Opeyemi Rebecca Akinajo[1,2,3‡], Giulia Scarpa[3‡], Claudia Hanson[1‡], Elin C. Larsson[1,4‡], Bosede Bukola Afolabi[2,5‡], Lenka Beňová[3☯]

1 Department of Global Public Health, Karolinska Institutet, Stockholm, Sweden, 2 Department of Obstetrics and Gynaecology, College of Medicine, University of Lagos/ Lagos University Teaching Hospital, Idi-Araba, Lagos, Nigeria, 3 Department of Public Health, Institute of Tropical Medicine, Antwerp, Belgium, 4 Department of Womens and Childrens Health, Karolinska Institutet, Stockholm, Sweden, 5 Centre for Clinical Trials, Research and Implementation Science, College of Medicine, University of Lagos, Idi-Araba, Lagos, Nigeria

☯ These authors contributed equally to this work
‡ ORA, ECL and BBA also contributed equally to this work
* ochuwa.babah@ki.se

## Abstract

The prevalence of anaemia in pregnancy differs across regions worldwide. Previous studies have reported associations between diet and anaemia. Dietary intake may be affected by regional factors like culture, food production and availability. However, in Nigeria, the association between dietary diversity and region in the context of anaemia prevalence among pregnant women is unclear. This study compared the prevalence of anaemia in pregnancy across regions in Nigeria and determined the association between dietary diversity and anaemia across the regions. It was across-sectional study of 1,525 pregnant women aged 15–49 screened for anaemia in Nigeria's Demographic and Health Survey 2018. The primary outcome was anaemia (haemoglobin concentration < 11g/dl, irrespective of trimester). The explanatory variable was minimum dietary diversity for women (MDD-W) defined as the consumption of at least five out of ten food groups on the day preceding the interview, stratified by region. Logistic regression analyses were used to determine the association between dietary diversity and anaemia in pregnancy by region. The prevalence of anaemia in pregnancy was 61.1% and it ranged from 55.2% in South-West to 71.1% in South-East region, p = 0.038. Less than half of pregnant women met the MDD-W requirement (45.8%). There was a significant crude association between MDD-W and anaemia, OR: 0.78 (95%CI: 0.60 - 0.99), which was lost when confounders were included, aOR: 0.85 (95%CI: 0.66-1.10). Compared to North-West region, anaemia

**Data availability statement:** The dataset analysed for this study is available from the Nigeria's DHS 2018 individual recode Stata dataset (.dta), Accession number: NGIR7BDT. ZIP.

**Funding:** The author(s) received no specific funding for this work.

**Competing interests:** The authors have declared that no competing interests exist.

in pregnancy was significantly higher in North Central region aOR:1.90 (1.14–3.16). The model with an interaction term between MDD-W and region was not a better fit for the data (LRtest p < 0.001) in multivariable model. In conclusion, the prevalence of anaemia in pregnancy is high in Nigeria and varies across regions, not only due to dietary diversity. Region is not an effect modifier of the association between MDD-W and anaemia.

## Plain language summary

Anaemia in pregnancy is a reduction in the red blood cells that carry oxygen within the body during pregnancy. It is a global health issue and is known to be among the top ten leading causes of maternal death. About six out of ten pregnant women in Nigeria have anaemia, but this varies from region to region. If not promptly diagnosed and treated, anaemia may harm the health of the mother and/or her baby. Previous studies have identified diet as a risk factor for anaemia in pregnancy. We are aware that women's diet is influenced by factors like type of foods available, cultural practices including food taboos, and the household's purchasing power, which in turn influences the quantity and quality of food consumed. These factors affect the dietary diversity and can affect whether pregnant women develop anaemia. This study investigated whether the diversity of food groups women consume could explain why the occurrence of anaemia differs across regions in Nigeria. We found that diet alone does not explain the regional differences in anaemia occurrence, suggesting that there are other factors which may better explain the difference in anaemia's occurrence across regions of Nigeria.

## Introduction

Anaemia in pregnancy is one of the non-communicable diseases targeted by the United Nations Sustainable Development Goal (SDG) 3 geared towards reducing maternal mortality [1]. In 2019, anaemia was estimated to have affected 32 million pregnant women worldwide [2]. Pregnant women are at higher risk of anaemia compared to non-pregnant women because of the physiological haemodilution that occurs during pregnancy and the demands of the growing foetus for micronutrients which play an important role in haematopeoisis [3]. Anaemia in pregnancy is associated with poor pregnancy outcomes, with adverse sequelae for mother and baby including an increased risk of preterm birth and postpartum haemorrhage, foetal growth restriction, leading to low birth-weight or intrauterine foetal death, and maternal death [4].

All countries in sub-Saharan Africa (SSA) except the Seychelles are low- or middle-income countries (LMICs) with a large proportion of the population living below the poverty line [5,6]. Poor populations are prone to a higher prevalence of

inadequate nutrient intake leading to health problems including anaemia [7]. The burden of anaemia in pregnancy is greatest in LMICs where prevalence is above 40% compared to high-income countries where the prevalence is below 20% [2]. The prevalence of anaemia in pregnancy has remained high despite routine iron supplementation, suggesting a need to shift focus towards adopting diverse strategies geared towards prevention [8]. These strategies support intervening in the preconception stage and also emphasise the need for health educational and nutritional intervention from the adolescent stage, considering that anaemia is prevalent in all age groups in females [9].

To make preventive strategies more effective, it is important to identify modifiable factors associated with anaemia. Some factors which have been identified include higher parity, later trimester of pregnancy, low socioeconomic status and inadequate nutrient intake [10,11]. However, the variation of such risk factors in Nigeria is understudied. Previous studies in Latin America, Luxembourg, and other LMICs have consistently reported an association between socioeconomic status and dietary intake; suggesting that those in lower socioeconomic positions consume fewer vegetables, fruits, seafoods including fish, and whole grains, but consume more legumes and with lower dietary diversity compared to people in higher socioeconomic position [12–14]. Dietary diversity is defined as "the number of different food groups consumed over a given reference period" [15].

Nigeria is a large country located in SSA, and with six geopolitical regions – three each in the Northern and Southern parts of the country [16]. These regions differ in terms of culture, socio-economic status, food production and food availability [17]. Facility-based studies have pointed to larger differences in the prevalence of anaemia in pregnancy ranging from 32.5% in Ogun state in the South-West region to 61.1% in Akwa Ibom in the South-South region [18–21]. Evidence suggests that dietary diversity in Nigeria is poor, with only half of reproductive aged women (15–49 years) and less than one-quarter of children have adequate dietary diversity [22]. There is paucity of studies from Nigeria exploring association between dietary diversity and the prevalence of anaemia in pregnancy.

A previous study in two states of Nigeria which examined specific food types and their association with anaemia in pregnancy found that the consumption of soybeans and edible kaolin clay was associated with an increased risk for iron deficiency anaemia among pregnant women, while the consumption of green leafy vegetables decreased the odds of anaemia [18]. An analysis of the Demographic and Health Survey (DHS) 2018 found an association between dietary diversity and anaemia among women of reproductive age – both pregnant and non-pregnant [23]. However, it is not known how disparities in dietary diversity (based on the minimum dietary diversity for women (MDD-W)) across regions may contribute to the differences in the prevalence of anaemia in pregnancy. There is a need to understand the impact of dietary diversity based on MDD-W on anaemia, and anaemia severity among pregnant women - overall and across regions. This study aimed to estimate and compare the prevalence of anaemia and its severity in pregnant women nationally and in the six regions of Nigeria. It also aimed to determine the association between MDD-W and the prevalence of anaemia among pregnant women, and whether this association varies across regions.

## Methods

### Ethics statement

The DHS received ethical approval for the survey protocol from the National Health Research Ethics Committee of Nigeria (NHREC) and the ICF Institutional Review Board before conducting the survey. The confidentiality of the study participants was assured during the survey; to ensure this, the data was de-identified [24]. Informed consent was obtained from all the study participants during the DHS. We obtained permission from The DHS Program (11th March 2024) to grant access to the database.

### Study design and data source

This was a cross-sectional study using data from the Nigeria DHS 2018 [24]. The data, which was deidentified, was accessed on 9th May 2024. The DHS is a household survey conducted between August and December 2018 and contains nationally and region-specific representative samples [24]. The DHS collected information on socio-economic factors of

the respondents and the households, nutritional status of reproductive aged women, maternal health, and other reproductive health issues.

## Study location

Nigeria is a sub-Saharan African country with a population of over 227 million as of 2023 [25]. It is bounded by the Republic of Niger in the North, by Cameroon and Chad Republic in the East, by Benin Republic in the West and by the Gulf of Guinea in the Atlantic Ocean in the South [26]. The country is divided into six geopolitical zones or regions [26]. It is a country with diverse ethnicity, with about 250 ethnic groups [26]. Its economy is one of the largest in sub-Saharan Africa [26].

## Study population

The unit of analysis is reproductive aged women who reported being pregnant at the time of the survey and who, after providing written informed consent, were screened for anaemia. Fig 1 gives details of the participants selection for this study. The survey used a multistage sampling technique to select a total 42,000 households; in one-third of the sampled households (14,000), reproductive-age women were screened for anaemia, and those pregnant women among this sub-sample were included in this study [24].

## Data collection

Data were collected on electronic devices by trained enumerators. For biomarkers which included haemoglobin concentration measurements, the data was first recorded on paper questionnaires and subsequently entered into the electronic database. Quality assurance strategies put in place before and during the survey included training and re-training of all the research team members. The enumerators were trained on how to enter data on the electronic platform, how to use the biomarker checklist and how to collect specimens for investigations. Anthropometric measurements suspected to be incorrect were flagged and re-measurement performed. Six quality control officers were nominated, and together with the state coordinators, they monitored the fieldwork to ensure compliance with the protocol and research ethics and to promptly identify issues as they occurred.

## Outcome variable

The primary outcome was presence of anaemia (yes/no). Anaemia, measured with Hemocue 301 device, was defined as haemoglobin concentration less than 11g/dl irrespective of the trimester of pregnancy [1]. Anaemia severity was graded as mild (10.0-10.9g/dl), moderate (7.0-9.9g/dl), or severe (<7.0g/dl) [27].

## Explanatory variables

Main exposure was MDD-W, which is a dichotomous (yes/no) indicator developed by Food and Agriculture Organization (FAO) [28]. MDD-W is an international measure that has been utilised in over 85 countries, capturing the number of food groups consumed on the preceding day and night [28]. A food group refers to a variety of foods that have similar nutritional constituents with similar value on health. From a total of ten food groups, e.g., grains and tubers, pulses like beans, nuts and seeds, green leafy vegetables, other vegetables, eggs, etc, women who consumed fewer than five food groups were considered not to have the minimum dietary diversity. According to FAO, MDD-W is a proxy indicator measuring populations' micronutrient adequacy based on self-reported information [28]. A list of the food in each of the ten food groups is presented in S1 Table.

## Effect modifier

Regions of the country were considered as potential effect modifiers; the six regions are North-West, North-East, North-Central, South-West, South-East and South-South, Fig 2.

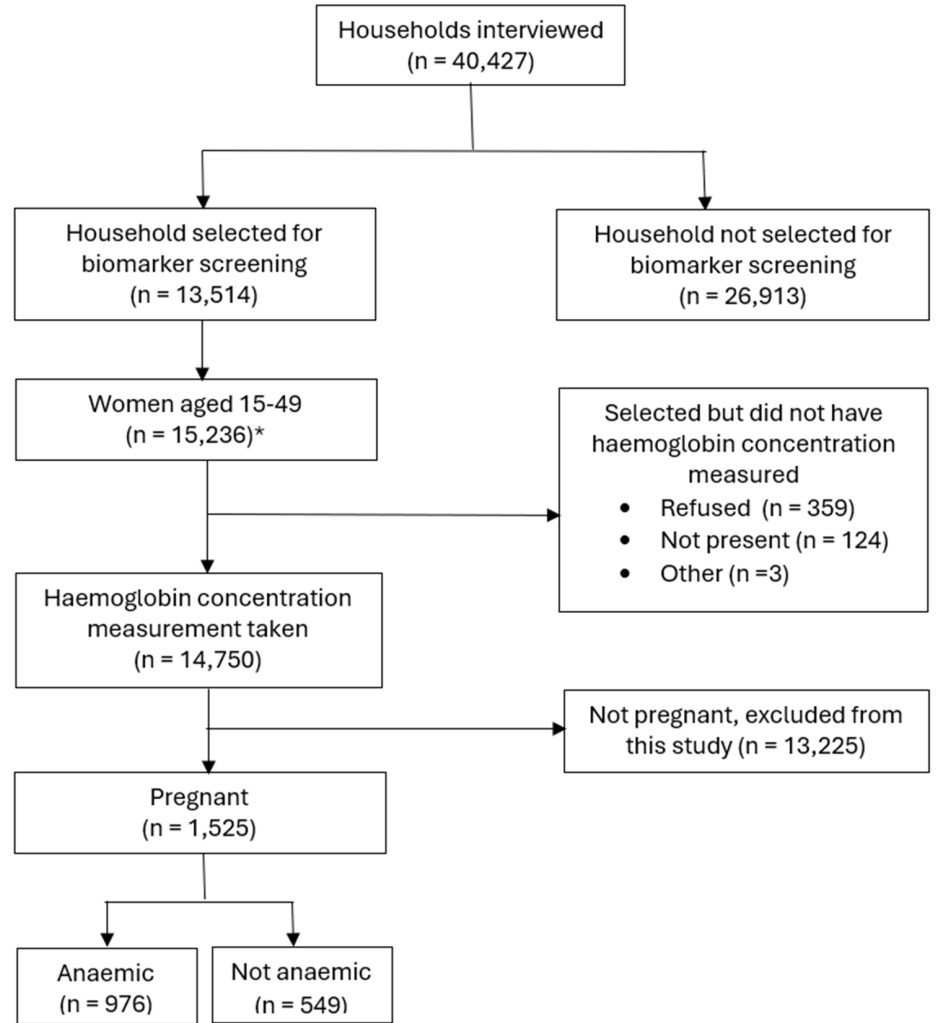

*All eligible pregnant women were included in the survey; multiple women per household were possible.*

**Fig 1. Flow chart of selection of participants for this study.**

## Other variables

Other variables of interest included age groups of the women at last birthday in years (15–20, 20–29, 30–39 or 40–49); trimester of pregnancy (first, second or third); place of residence (rural/urban); religion (Christianity, Islam, or traditional/other); ethnicity (Igbo, Yoruba, Hausa, or other ethnic groups); marital status at time of survey (currently married/cohabiting or not (divorced, separated, widowed, or never married); employment status at the time of survey (employed or unemployed); highest level of education (no formal education, primary, secondary or higher); number of children ever born, that is, parity (0, 1, 2–4, and 5 or more); and household wealth quintile (poorest, poorer, middle, richer and richest) [29].

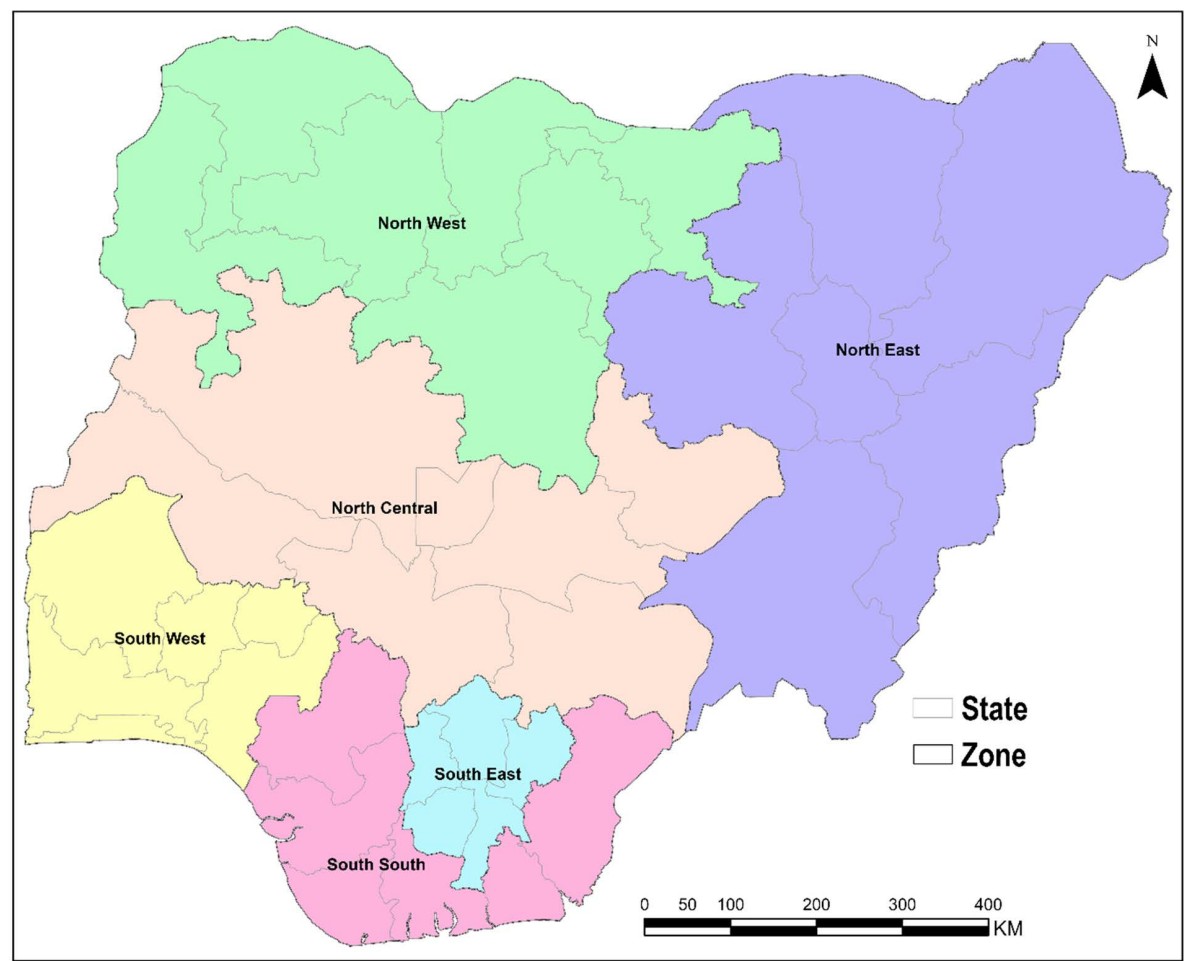

Source: This map was plotted by the author in ArcGIS Pro V 3.3 (ESRI, Redlands, CA, USA) based on boundaries from GRID3 under a CC BY 4.0 License at: https://data.grid3.org/datasets/GRID3::grid3-nga-operational-lga-boundaries/about

**Fig 2. Map of Nigeria showing six regions.**

## Analysis

Data were analyzed using Stata (StataCorp (2023) Stata Statistical Software: Release 18. StataCorp LLC, College Station, TX). Descriptive statistics were done. Continuous variables that were normally distributed were presented as mean with 95% confidence interval (95%CI) while those that deviated from normality were presented as median and interquartile range. Categorical variables were presented as percentages and 95%CI. Prevalence of anaemia (overall and by severity) for each region was presented as percentages with 95%CI and compared across the regions using Chi-square test. Haemoglobin concentrations by region were presented as mean and 95%CI and comparisons across regions were made using Student's t-test. Comparisons of MDD-W and food groups consumed by regions were done using Chi-square test and visually displayed using heat maps.

We used logistic regression to examine the association between MDD-W and the binary variable anaemia. Bivariate models were constructed to determine the independent effect of each covariate on anaemia. Thereafter, a multivariable

logistic regression was conducted adjusting for relevant covariates. A forward stepwise selection method was used. A p-value threshold of <0.10 was used to select variables that were retained in the model. The covariates considered were the main explanatory variables – MDD-W and region, and those known or have been found in previous studies to be associated with anaemia or are potential confounders. Finally, an interaction term was fitted to determine if there was an interaction between MDD-W and region. Postestimation was done with likelihood ratio test comparing the model with and without the interaction term to determine which of the two models was better. The Akaike Information Criterion (AIC) and Bayesian Information Criterion (BIC) for both models are presented. Further analysis was conducted to determine the association between specific food groups and anaemia in pregnancy. All analyses were adjusted for clustering, stratification, and sampling weights. Complete case analysis was done; no imputation technique was applied to handle missing data. Observations with missing values in any of the variables were excluded from analysis and the extent of missingness was reported.

## Results

A total of 15,236 reproductive aged women were invited for anaemia screening. Of the women, 486 did not have anaemia screening done because they either refused screening (n = 359), did not report for screening (n = 124) or gave other reasons (n = 3). A total of 14,750 were screened for anaemia, of which only 1,525 were pregnant women, Fig 1. All the 1,525 eligible pregnant women who participated in the survey and had a valid haemoglobin concentration measurement were included in this study. Table 1 shows details of the sociodemographic profile of women in the analysis sample. Largest percentage of them were within the age group 20–29, 52.0%; parity 2–4, 43.1%; and rural residents 61.0% (95%CI: 58.1-63.8%).The distribution of the pregnant women varied across region, ranging from 8.5% (95%CI: 7.2-9.9%) of the sample in the South-South to 36.4% (95%CI: 33.4-39.5%) of the sample in the North-West.

The overall prevalence of anaemia among pregnant women was 61.1% (95%CI: 58.0-64.2%), Table 2, with significant differences across the six regions (p = 0.038). South-East region had the highest prevalence of 71.1% (95%CI: 62.3-78.7%), while South-West region had the lowest prevalence of 55.2% (95%CI: 43.8-66.1%). However, the mean haemoglobin concentration among pregnant women was not statistically significantly different across regions (p = 0.422). In the sub-population of pregnant women with anaemia in pregnancy, the mean haemoglobin concentration, 9.5 (95%CI: 9.4-9.6) g/dl, was also similar across regions (p = 0.689).

Looking at effect of trimester on anaemia and haemoglobin concentration, Table 2 shows that the prevalence of anaemia increases significantly as trimester advances, with lowest prevalence of 44.7 (95%CI: 39.1 - 50.4) in the first trimester and highest prevalence of 69.4 (95%CI: 64.5 - 74.0) in the third trimester (p < 0.001). The mean haemoglobin concentration drops significantly as trimester advances with highest values in the first trimester and lowest values in the third trimester in both general population of pregnant women and in sub-group of pregnant women with anaemia (p < 0.001 and p = 0.02 respectively).

Fig 3 shows the prevalence of anaemia in pregnancy by severity in each region. The prevalence of severe anaemia was highest in the South-South (11.0%) and lowest in the South-West (1.0%).

Fig 4 presents a heat map showing MDD-W by region among pregnant women (n = 1,502). Across the regions, most women consumed food from 3-5 food groups. MDD-W (five or more food groups) was least achieved among pregnant women in the North-Central region (p < 0.001).

S1 Fig shows the pattern of consumption by exact food groups among pregnant women, by region. The consumption of foods in group 1 which comprised grains, white tubers, plantain, and roots, was significantly lower in South-West region where the prevalence of anaemia in pregnancy was lowest compared to other regions with higher prevalence.

The binary logistic regression analysis showed that MDD-W was associated with anaemia in pregnancy (Table 3). Pregnant women achieving MDD-W had 22% lower crude odds of anaemia compared to those who did not (95%CI: 0.60-0.99). After adjustment for confounders, the odds of anaemia in pregnant women who achieved MDD-W remained lower, but the association was no longer statistically significant; aOR: 0.85 (0.66-1.10), p = 0.219. On the other hand, region had a significant adjusted effect on the odds of anaemia in pregnancy. Pregnant women in the North-Central region had 90% higher

**Table 1. Sociodemographic characteristics of analysis sample of pregnant women from Nigeria DHS 2018 (n = 1,525).**

| Sociodemographic characteristic | All pregnant women | | Anaemic (Hb < 11g/dl) | | Chi-square p-value |
|---|---|---|---|---|---|
| | n = 1,525 | % (95%CI) | n = 976 | % (95%CI) | |
| **Region** | | | | | |
| North-Central | 290 | 14.7 (13.1 - 16.6) | 199 | 16.8 (14.7 - 19.1) | **0.038** |
| North-East | 305 | 17.8 (15.9 - 19.9) | 184 | 16.3 (14.6 - 18.2) | |
| North-West | 458 | 36.4 (33.4 - 39.5) | 293 | 35.7 (33.1 - 38.4) | |
| South-East | 179 | 10.1 (8.8 - 11.6) | 129 | 11.8 (10.3 - 13.4) | |
| South-South | 137 | 8.5 (7.2 - 9.9) | 81 | 8.2 (6.7 - 10.0) | |
| South-West | 156 | 12.5 (10.7 - 14.5) | 90 | 11.3 (9.6 - 13.2) | |
| **Age group (years)** | | | | | |
| 15 – 19 | 150 | 10.6 (8.9 - 12.5) | 99 | 11.0 (9.1 - 13.3) | 0.397 |
| 20 – 29 | 772 | 52.0 (49.2 - 54.9) | 488 | 51.3 (47.8 - 54.7) | |
| 30 – 39 | 514 | 32.2 (29.4 - 35.0) | 340 | 33.2 (30.0 - 36.6) | |
| 40 – 49 | 89 | 5.2 (4.1 - 6.6) | 49 | 4.5 (3.2 - 6.2) | |
| **Trimester of pregnancy** | | | | | |
| First | 444 | 31.0 (28.1 - 34.0) | 217 | 22.6 (19.8 - 25.7) | **<0.001** |
| Second | 562 | 35.3 (32.3 - 38.3) | 382 | 39.0 (35.6 - 42.5) | |
| Third | 519 | 33.7 (30.8 - 36.8) | 377 | 38.4 (34.8 - 42.1) | |
| **Parity** | | | | | |
| 0 | 208 | 14.0 (12.0 - 16.2) | 135 | 14.5 (12.3 - 17.2) | 0.597 |
| 1 | 275 | 19.1 (16.6 - 21.9) | 174 | 17.9 (15.1 - 21.0) | |
| 2 – 4 | 656 | 43.1 (40.3 - 46.0) | 422 | 44.1 (40.6 - 47.6) | |
| 5 or higher | 386 | 23.8 (21.3 - 26.4) | 245 | 23.5 (20.7 - 26.6) | |
| **Marital status** | | | | | |
| Not married | 43 | 2.7 (2.0 - 3.7) | 34 | 3.6 (2.5 - 5.0) | **0.013** |
| Married/cohabiting | 1,482 | 97.3 (96.3 - 98.0) | 942 | 96.4 (95.0 - 97.5) | |
| **Ethnicity** | | | | | |
| Hausa | 487 | 38.0 (34.8 - 41.4) | 312 | 37.7 (34.4 - 41.1) | 0.117 |
| Igbo | 205 | 12.2 (10.5 - 14.2) | 145 | 14.2 (12.1 - 16.7) | |
| Yoruba | 154 | 11.4 (9.8 - 13.3) | 90 | 10.8 (9.0 - 12.9) | |
| Other/don't know | 679 | 38.3 (35.3 - 41.4) | 429 | 37.3 (34.1 - 40.6) | |
| **Residence** | | | | | |
| Urban | 540 | 39.0 (36.2 - 41.9) | 314 | 35.6 (32.9 - 38.3) | **0.008** |
| Rural | 985 | 61.0 (58.1 - 63.8) | 662 | 64.4 (61.7 - 67.1) | |
| **Highest level of education** | | | | | |
| No formal education | 666 | 43.8 (40.8 - 46.9) | 452 | 45.9 (42.4 - 49.4) | **<0.001** |
| Primary | 216 | 13.4 (11.6 - 15.5) | 147 | 15.4 (13.0 - 18.2) | |
| Secondary | 520 | 35.0 (32.2 - 38.0) | 321 | 33.3 (30.1 - 36.6) | |
| Higher | 123 | 7.8 (6.4 - 9.5) | 56 | 5.4 (4.0 - 7.2) | |
| **Religion** | | | | | |
| Christianity | 611 | 37.8 (34.9 - 40.9) | 375 | 36.6 (33.6 - 39.8) | 0.256 |
| Islam | 902 | 61.6 (58.6 - 64.6) | 596 | 63.0 (59.8 - 66.1) | |
| Traditional/other | 6 | 0.5 (0.2 - 1.2) | 5 | 0.4 (0.1 - 1.0) | |

*(Continued)*

**Table 1.** (Continued)

| Sociodemographic characteristic | All pregnant women | | Anaemic (Hb < 11g/dl) | | Chi-square p-value |
|---|---|---|---|---|---|
| | n = 1,525 | % (95%CI) | n = 976 | % (95%CI) | |
| Wealth quintile | | | | | |
| Poorest | 344 | 21.1 (18.5 - 24.0) | 243 | 23.0 (20.0 - 26.2) | 0.012 |
| Poorer | 360 | 22.5 (19.9 - 25.4) | 240 | 24.3 (21.1 - 27.7) | |
| Middle | 338 | 22.1 (19.4 - 25.2) | 225 | 22.7 (19.7 - 26.1) | |
| Richer | 273 | 18.7 (16.1 - 21.5) | 162 | 17.1 (14.5 - 20.0) | |
| Richest | 210 | 15.6 (13.3 - 18.1) | 106 | 13.0 (10.4 - 16.0) | |
| | | Mean (SD) | | Mean (SD) | |
| Weight (kilogramme) | 1,525 | 59.9 (58.4 - 61.5) | 976 | 58.7 (57.9 - 59.5) | – |
| Height (metres) | 1,525 | 1.59 (1.57 - 1.60) | 976 | 1.58 (1.576 - 1.58) | – |
| Body mass index (kg/m²) | 1,525 | 23.7 (23.4 - 23.9) | 976 | 23.4 (23.1 - 23.7) | – |

*Chi-square p-value compared anaemia prevalence across levels/sub-groups of each sociodemographic characteristic.*

**Table 2.** Prevalence of anaemia and mean haemoglobin concentration levels among pregnant women in Nigeria (n = 1,525).

| Variable | Total | Prevalence of anaemia among pregnant women, %(95%CI) (n = 1,525) | Chi square p-value | Haemoglobin concentration among all women, g/dl Mean (95%CI) (n = 1,525) | t-test p-value | Number of anaemic women in sample, n | Haemoglobin concentration among anaemic women, g/dl Mean (95%CI) (n = 976) | t-test p-value |
|---|---|---|---|---|---|---|---|---|
| **Regions** | | | | | | | | |
| North-Central | 290 | 69.4 (62.9 - 75.2) | 0.038 | 10.4 (10.1 - 10.6) | 0.422 | 199 | 9.6 (9.4 - 9.8) | 0.689 |
| North-East | 305 | 56.0 (49.2 - 62.5) | | 10.7 (10.4 - 10.9) | | 184 | 9.6 (9.4 - 9.8) | |
| North-West | 458 | 59.9 (54.6 - 65.1) | | 10.5 (10.3 - 10.7) | | 293 | 9.4 (9.3 - 9.6) | |
| South-East | 179 | 71.1 (62.3 - 78.7) | | 10.1 (9.9 - 10.4) | | 129 | 9.5 (9.3 - 9.7) | |
| South-South | 137 | 59.2 (47.4 - 70.1) | | 10.4 (10.0 - 10.8) | | 81 | 9.5 (9.1 - 9.8) | |
| South-West | 156 | 55.2 (43.8 - 66.1) | | 10.8 (10.4 - 11.3) | | 90 | 9.7 (9.5 - 9.9) | |
| **Trimesters** | | | | | | | | |
| First trimester | 444 | 44.7 (39.1 - 50.4) | <0.001 | 11.2 (10.9 - 11.4) | <0.001 | 217 | 9.7 (9.5 - 9.9) | 0.002 |
| Second trimester | 562 | 67.6 (63.4 - 71.5) | | 10.3 (10.1 - 10.4) | | 382 | 9.5 (9.4 - 9.6) | |
| Third trimester | 519 | 69.4 (64.5 - 74.0) | | 10.2 (10.0 - 10.3) | | 377 | 9.4 (9.3 - 9.5) | |
| **Overall** | 1,525 | 61.1 (58.0 - 64.2) | | 10.5 (10.4 - 10.6) | | 976 | 9.5 (9.4 - 9.6) | |

*p-value compared the prevalence or mean haemoglobin concentration across levels/subgroups of each variables.*

odds for anaemia in pregnancy, aOR: 1.90 (1.13-3.16), compared to the North-West region. The adjusted odds of anaemia in pregnancy were more than double in the second and third trimesters of pregnancy compared to the first, p < 0.001. Not being married increased the odds of anaemia, compared to those married/cohabiting, and women with tertiary education had 64% lower odds of anaemia in pregnancy (p = 0.002) compared to those with no formal education. Postestimation comparison of the full model with an interaction term between region and MDD-W versus the model without the interaction term showed that the data were consistent with no evidence of effect modification, p = 0.154. The model with the interaction term had AIC = 1,877 and BIC = 2,047 while the model without the interaction term had AIC = 1,875 and BIC = 2,019 (Table 3).

Within each of the six regions, there was no association between MDD-W and anaemia among pregnant women (S2 Table).

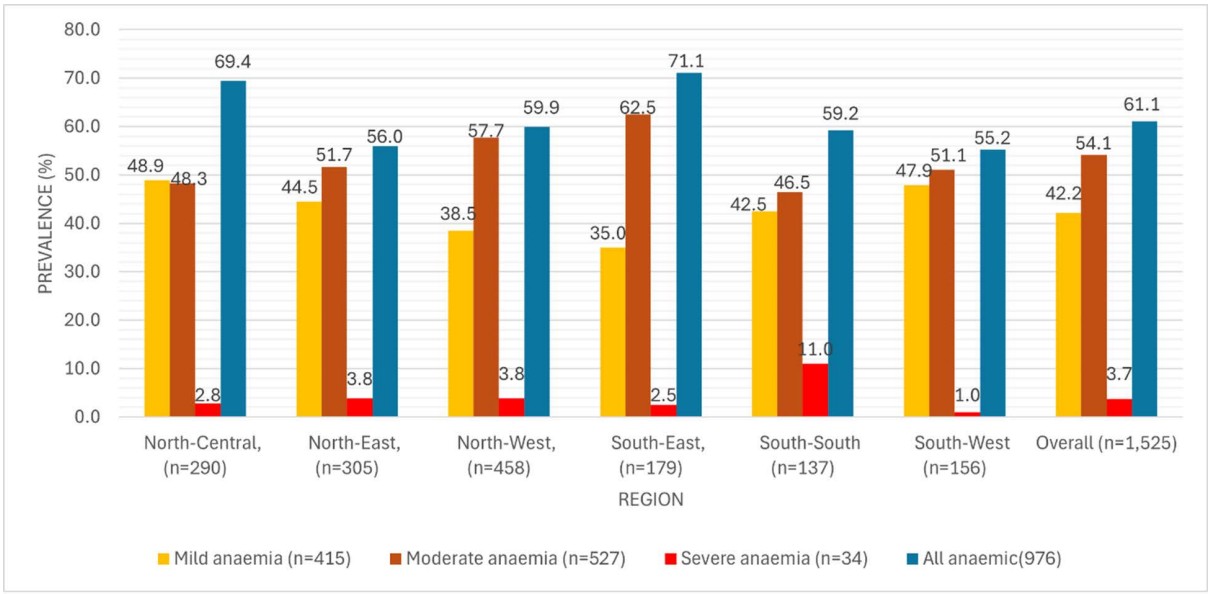

Mild anaemia was defined as haemoglobin concentration between 10.0g/dl and 10.9g/dl, moderate anaemia as haemoglobin concentration between 7.0g/dl and 9.9g/dl, and severe anaemia as haemoglobin concentration of less than 7.0g/dl.
Yellow, brown and red bars represent prevalence of anaemia by level of severity among pregnant women with anaemia (n = 976).
Blue bar represents prevalence of anaemia among all pregnant women (n = 1,525).

**Fig 3. Prevalence of anaemia by severity and among all pregnant women across six regions in Nigeria, among anaemic pregnant women.**

| Region | Number of food groups consumed the previous day (dietary diversity score) | | | | | | | | | | | 5 or more | Mean MDD score |
|---|---|---|---|---|---|---|---|---|---|---|---|---|---|
| | None | One | Two | Three | Four | Five | Six | Seven | Eight | Nine | Ten | | |
| NIGERIA | 0.1 | 2.3 | 9.6 | 20.0 | 24.8 | 21.7 | 11.4 | 6.2 | 2.5 | 0.6 | 0.9 | 45.8 | 4.46 |
| North-Central | 0.2 | 7.4 | 13.3 | 22.4 | 19.6 | 19.0 | 11.2 | 5.4 | 1.5 | 0.0 | 0.0 | 37.1 | 3.92 |
| North-East | 0.0 | 2.4 | 3.7 | 16.1 | 30.0 | 24.3 | 15.6 | 5.5 | 2.0 | 0.4 | 0.0 | 47.9 | 4.52 |
| North-West | 0.0 | 1.3 | 6.9 | 19.5 | 24.4 | 19.7 | 13.6 | 9.0 | 4.0 | 1.4 | 0.2 | 47.9 | 4.61 |
| South-East | 0.0 | 1.7 | 8.2 | 22.2 | 24.6 | 20.8 | 7.8 | 4.5 | 2.9 | 0.8 | 6.5 | 43.3 | 4.61 |
| South-South | 0.0 | 0.0 | 6.0 | 20.5 | 31.9 | 22.6 | 10.7 | 2.0 | 5.7 | 0.6 | 0.0 | 41.6 | 4.43 |
| South-West | 0.2 | 0.2 | 12.3 | 19.4 | 16.1 | 19.8 | 25.1 | 5.1 | 0.6 | 0.6 | 0.6 | 51.7 | 4.48 |
| p-value | | | | | | | | | | | | | 0.032 |

Colour code:   ◯ 0-10%   ◯ 10-20%   ◯ 20-30%   ● >30-40%   ● >40-50%   ● >50%

Figures presented proportion of pregnant women who achieved various dietary diversity scores 0-10; reported as percentage of total population of pregnant women per region.

p-value compared the mean dietary diversity score across the six geographical regions within the country.

Food group: 1 – grains, white roots, tubers, plantain; 2 – pulses (beans, peas, lentils); 3 – nuts and seeds; 4 – dairy (milk and milk products); 5 – meat, poultry, fish, small insects; 6 – eggs; 7 – dark green leafy vegetables; 8 – vitamin A rich fruits/vegetables like mango, pawpaw, watermelon, tomato, carrots; 9 – other vegetables, 10 – other fruits.

**Fig 4. Heat map showing number of food groups consumed by pregnant women in the six regions in Nigeria (n =1,502).**

**Table 3. Crude and adjusted logistic regressions models predicting anaemia in pregnancy in Nigeria.**

| Predictors | Crude | | | Multivariable n = 1,502 | |
|---|---|---|---|---|---|
| | OR (95%CI) | p-value | LR p-value | aOR (95%CI) | Wald p-value |
| MDD-W | n = 1,502 | | | | |
| Not achieved (<5 groups) | 1.00 | – | 0.049 | 1.00 | – |
| Achieved (≥5 groups) | 0.78 (0.60 - 0.99) | **0.049** | | 0.85 (0.66-1.10) | 0.219 |
| Region | n = 1,525 | | | | |
| North-Central | 1.52 (1.06 - 2.18) | **0.024** | 0.012 | 1.90 (1.14-3.16) | **0.014** |
| North-East | 0.85 (0.60 - 1.20) | 0.357 | | 0.95 (0.62-1.47) | 0.816 |
| North-West | 1.00 | – | | 1.00 | – |
| South-East | 1.65 (1.05 - 2.59) | **0.031** | | 1.34 (0.42-4.23) | 0.622 |
| South-South | 0.97 (0.58 - 1.64) | 0.912 | | 1.67 (0.81-3.47) | 0.167 |
| South-West | 0.82 (0.50 - 1.37) | **0.012** | | 1.26 (0.54-2.93) | 0.587 |
| Age group | n = 1,525 | | | | |
| 15 – 20 | 1.00 | – | 0.394 | 1.00 | – |
| 20 – 29 | 0.86 (0.58-1.28) | 0.456 | | 0.99 (0.60-1.65) | 0.973 |
| 30 – 39 | 0.98 (0.63-1.51) | 0.909 | | 1.31 (0.66-2.60) | 0.436 |
| 40 – 49 | 0.61 (0.32-1.17) | 0.138 | | 0.70 (0.30-1.63) | 0.404 |
| Trimester | n = 1,525 | | | | |
| First | 1.00 | – | <0.001 | 1.00 | – |
| Second | 2.58 (1.91 - 3.47) | **<0.001** | | 2.46 (1.81-3.34) | **<0.001** |
| Third | 2.81 (2.06 - 3.83) | **<0.001** | | 2.88 (2.09-3.96) | **<0.001** |
| Parity | n = 1,525 | | | | |
| 0 | 1.00 | – | 0.678 | 1.00 | – |
| 1 | 0.77 (0.48-1.23) | 0.276 | | 0.95 (0.58-1.56) | 0.852 |
| 2 – 4 | 0.95 (0.67-1.36) | 0.791 | | 0.99 (0.60-1.65) | 0.983 |
| ≥5 | 0.88 (0.59-1.30) | 0.520 | | 0.71 (0.38-1.33) | 0.286 |
| Marital status | n = 1,525 | | | | |
| Not married | 2.66 (1.19 - 5.94) | **0.017** | 0.017 | 2.59 (1.10-6.09) | **0.029** |
| Married/cohabiting | 1.00 | – | | 1.00 | – |
| Ethnicity | n = 1,525 | | | | |
| Hausa | 1.00 | – | 0.091 | 1.00 | – |
| Igbo | 1.59 (1.02-2.48) | **0.042** | | 2.21 (0.72-6.79) | 0.168 |
| Yoruba | 0.89 (0.57-1.40) | 0.606 | | 1.16 (0.50-2.70) | 0.728 |
| Other | 0.95 (0.71-1.28) | 0.747 | | 0.87 (0.57-1.31) | 0.504 |
| Place of residence | n = 1,525 | | | | |
| Urban | 0.69 (0.53 - 0.91) | **0.008** | 0.009 | 0.79 (0.57-1.10) | 0.168 |
| Rural | 1.00 | – | | | |
| Level of education | n = 1,525 | | | | |
| None | 1.00 | – | <0.001 | 1.00 | – |
| Primary | 1.32 (0.91 - 1.92) | 0.140 | | 1.30 (0.86-1.99) | 0.216 |
| Secondary | 0.78 (0.58 - 1.04) | 0.085 | | 0.69 (0.46-1.03) | 0.068 |
| Tertiary or higher | 0.41 (0.26 - 0.64) | **<0.001** | | 0.36 (0.18-0.69) | **0.002** |
| Religion | n = 1,519 | | | | |

*(Continued)*

**Table 3.** (Continued)

| Predictors | Crude | | | Multivariable n = 1,502 | |
|---|---|---|---|---|---|
| | OR (95%CI) | p-value | LR p-value | aOR (95%CI) | Wald p-value |
| Christianity | 0.87 (0.66-1.14) | 0.315 | 0.206 | | |
| Islam | 1.00 | – | | | |
| Traditional/other | 0.43 (0.15-1.23) | 0.115 | | | |
| Wealth quintile | n = 1,525 | | | | |
| Poorest | 1.18 (0.80-1.73) | 0.411 | 0.018 | 1.49 (0.92-2.39) | 0.104 |
| Poorer | 1.14 (0.80-1.63) | 0.468 | | 1.25 (0.84-1.88) | 0.267 |
| Middle | 1.00 | – | | 1.00 | – |
| Richer | 0.75 (0.48-1.16) | 0.199 | | 0.93 (0.61-1.43) | 0.743 |
| Richest | 0.62 (0.40-0.96) | **0.032** | | 0.87 (0.51-1.48) | 0.598 |

OR – crude odds ratio; aOR – odds ratio adjusted for sociodemographic and obstetric characteristics; 95%CI – 95% confidence interval; LR – likelihood ratio.

Hosmer and Lemeshow p-value for multivariable model = 0.923; Tolerance = 0.892; Variance inflation factor, (VIF) = 1.12.

# Discussion

This study examined the association between MDD-W and anaemia in pregnancy, and whether this association varied across regions in Nigeria using recent nationally representative data. The prevalence of anaemia in pregnancy was high at 61%, with statistically significant variation across regions. Almost half of the pregnant women in Nigeria did not meet the requirements for minimum dietary diversity. While pregnant women achieving the MDD-W of five out of ten food groups had 15% lower odds of anaemia, this association remained in the same direction but was not statistically significant after the inclusion of confounders.

The prevalence of anaemia in pregnancy in Nigeria far exceeds previously reported prevalences by WHO across regions globally, including that in South-East Asia (48%) and SSA (46%) [2]. Comparing countries, the prevalence of anaemia in pregnancy in Nigeria is higher than that in India which had the highest prevalence in SE Asia of 50%, and is on level with Mali in Africa with the highest prevalence of 59% [2]. These differences across global regions may be related to the pattern of food consumption or food availability [30,31]. Anaemia in pregnancy has been linked to poor dietary intake as many cases have been associated with micronutrient deficiencies such as iron, folate, vitamin B12 and zinc deficiencies. In a facility-based study in two states of Nigeria, we found a prevalence of 41% iron deficiency among pregnant women with moderate or severe anaemia [18].

On specific food types, previously, we found that consumption of soybeans increased the risk of iron deficiency anaemia while consumption of green leafy vegetables significantly lowered the risk [18]. In this study, we observed that most of our pregnant women did not consume foods in group 6 which comprised eggs and foods in group 8 which comprised vitamin A-rich fruits and vegetables like mango, pawpaw, watermelon, tomato and carrots. This might be due to seasonal unavailability of the foods and can have an impact on dietary diversity and thus the prevalence of anaemia in pregnancy. Previous studies found that consumption of eggs, which are known to contain some amounts of heme and non-heme iron, improve haemoglobin concentration and serum ferritin levels [32,33]. Vitamin A deficiency has also been associated with anaemia in pregnancy [34,35]. Vitamin A plays a role in erythropoiesis by facilitating iron mobilization. In a state of deficiency, it causes an impairment in iron mobilisation from the liver and spleen, making it unavailable for erythropoiesis resulting in anaemia [36].

Fewer than half of the pregnant women in this study achieved the MDD-W, which exemplifies the burden of potential poor dietary intake, mainly in micronutrients, during pregnancy in Nigeria. Across the six regions in Nigeria, most women consumed foods from between three to five groups. Aside from the variation in socioeconomic status and food availability across regions, differences in culture, lifestyle and religion influence what people eat and may result in malnutrition due to restrictions in dietary intake predisposing to development of anaemia [37]. This may contribute to the differences in the prevalence of anaemia across regions. However, religion did not have an effect on the risk of anaemia in pregnancy in this study.

Furthermore, aside from the MDD-W and region, we found that the risk of anaemia increases as pregnancy progresses. This is an expected finding, likely as a result of the expansion of the plasma volume during pregnancy which results in physiological haemodilution predisposing to the occurrence of anaemia [38]. In addition, sociodemographic factors including not being married and no formal education were associated with higher odds of anaemia. Unmarried women had higher odds of anaemia compared to women who were married; this may be because of the lack of social and financial support which a partner is likely to offer in the Nigerian context [39,40]. Wealth has been found to be associated with risk for anaemia in previous studies [41–43]. The odds for anaemia was found to be lower with increasing household wealth index in this study in the unadjusted model, but the association was lost in the adjusted model. The latter may be due to a distortion of potential association because of residual confounding. The association between education and anaemia in pregnancy is not surprising because education is known to improve health literacy and promote knowledge-seeking behaviour in health-related aspects [44,45]. Based on our findings, the effect of education is unlikely to be due to financial means, because we adjusted for wealth in our analysis.

Overall, we found that dietary diversity alone was insufficient to explain the differences in the prevalence of anaemia in pregnancy across regions. The observed differences across regions might be explained by other factors aside from dietary differences. Future research should take into consideration non-pregnant women and also consider the seasonal availability of foods which might have an impact on food intake and nutritional status including anaemia in both pregnant and non-pregnant women. Considering that only less than 50% of pregnant women with anaemia might have underlying iron-deficiency, it is possible that non-iron-deficient anaemia might have a different association with food intake. It will thus be beneficial to determine in future research the association between food intake and non-iron-deficient anaemia.

This study will impact public health care in prioritising regions, especially the North-Central and South-East regions of Nigeria, for provision of health care intervention targeted at the less educated women in second and third trimesters of their pregnancy. There is a need to improve access to antenatal care and promote micronutrient adequacy for this category of pregnant women.

## Strengths and limitations

This study is population-based, with a sample which is representative of the country and thus provides valid insight into how MDD-W might affect differences in the prevalence of anaemia in pregnancy. However, while some studies have linked patterns of dietary intake and dietary diversity to the risk of anaemia [46,47], we observed a loss of statistical significance in the association between MDD-W and anaemia in pregnancy when confounders were considered. Though the association remained in the hypothesised direction, insufficient sample size might be one reason for the loss of power to detect it.

In DHS data, MDD-W is based on self-reported information and hence prone to recall bias. Although MDD-W is a proxy for micronutrient adequacy at the population level, it is based on individual data; hence in this study it was used as such to provide an insight into dietary diversity of women in Nigeria. Also, the method used to collect the MDD-W, does not allow the quantification of dietary consumption (e.g., size of portions); an important aspect to consider in dietary research. Moreover, it is important to consider when interpreting these findings that daily dietary intake is subject to fluctuations over time (before and during pregnancy), and seasonality of food availability.

In addition, data on iron and other micronutrient supplement intake, which is often offered and used by pregnant women, was not available in the DHS data. The aggregation of states within regions with differences in culture, food availability and immigration are potential confounding factors unadjusted for in this study.

## Conclusion

The prevalence of anaemia in pregnancy is high in Nigeria and varies across regions, and it is highest in the South-East. There seems to be an association between dietary diversity and anaemia in pregnancy which was no longer evident when sociodemographic factors were controlled for. Region does not differentiate the effect of dietary diversity on anaemia in pregnancy, but it may distort the association between them. Although, sociodemographic and obstetric factors like being single, being in the second and third trimester of pregnancy and having lower level of education were associated with increased risk of anaemia in pregnancy, our findings suggest that there are likely still other factors such as food security in terms of availability and accessibility and the seasonality of food availability which we could not account for in this study, including genetic haemoglobin disorders, maternal and community characteristics which might better explain why the disparity in the prevalence of anaemia in pregnancy across regions. These factors should be considered when developing or adapting public health strategies addressing anaemia in pregnant women.

## Supporting information

**S1 Table: List of food groups used to determine minimum dietary diversity for the pregnant women.**
(DOCX)

**S2 Table. Association between dietary diversity and anaemia among pregnant women nationally and across the six regions in Nigeria.**
(DOCX)

**S1 Fig. Pattern of food groups consumption by pregnant women across the six regions in Nigeria.**
(TIF)

## Acknowledgments

We acknowledge the Demographic and Health Surveys Program for granting us access to the DHS data analysed for this study and to all pregnant women who participated in the survey. We are also grateful to the facilitators and coaches of the DHS Summer Course 2024 organized at the Institute of Tropical Medicine, Antwerp, Belgium which was attended by the first (OAB) and third (OA) authors. Appreciations to Peter M. Macharia of Institute of Tropical Medicine, Antwerp, Belgium who helped create the map of Nigeria on ArcGIS Pro V 3.3.

## Author contributions

**Conceptualization:** Ochuwa Adiketu Babah, Diana Sagastume, Opeyemi Rebecca Akinajo, Lenka Beňová.

**Data curation:** Ochuwa Adiketu Babah.

**Formal analysis:** Ochuwa Adiketu Babah, Diana Sagastume, Giulia Scarpa, Lenka Beňová.

**Methodology:** Ochuwa Adiketu Babah, Diana Sagastume, Giulia Scarpa, Lenka Beňová.

**Project administration:** Ochuwa Adiketu Babah, Diana Sagastume.

**Supervision:** Diana Sagastume, Claudia Hanson, Elin C. Larsson, Bosede Bukola Afolabi, Lenka Beňová.

**Writing – original draft:** Ochuwa Adiketu Babah.

**Writing – review & editing:** Ochuwa Adiketu Babah, Diana Sagastume, Opeyemi Rebecca Akinajo, Giulia Scarpa, Claudia Hanson, Elin C. Larsson, Bosede Bukola Afolabi, Lenka Beňová.

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
