## [Decision Letter · Decision Letter 0]

28 Jan 2025

PGPH-D-24-02093

Dietary diversity insufficiently explains differences in prevalence of anaemia in pregnancy across regions in Nigeria: a secondary analysis of Demographic and Health Survey 2018

Dear Dr. Ochuwa Adiketu Babah, 

Thank you for submitting your manuscript to PLOS Global Public Health. After careful consideration, we feel that it has merit but does not fully meet PLOS Global Public Health’s publication criteria as it currently stands. Therefore, we invite you to submit a revised version of the manuscript that addresses the points raised during the review process.

We look forward to receiving your revised manuscript.

Kind regards,

Aditi Apte, MD PhD

Academic Editor

Journal Requirements:

2. We do not publish any copyright or trademark symbols that usually accompany proprietary names, eg (R), (C), or TM  (e.g. next to drug or reagent names). Please remove all instances of trademark/copyright symbols throughout the text, including ®  on Figure 1.

3. Please provide an Author Summary. This should appear in your manuscript between the Abstract (if applicable) and the Introduction, and should be 150–200 words long. The aim should be to make your findings accessible to a wide audience that includes both scientists and non-scientists. Sample summaries can be found on our website under Submission Guidelines:

https://journals.plos.org/globalpublichealth/s/submission-guidelines#loc-parts-of-a-submission.

4. Figure 1: please (a) provide a direct link to the base layer of the map (i.e., the country or region border shape) and ensure this is also included in the figure legend; and (b) provide a link to the terms of use / license information for the base layer image or shapefile. We cannot publish proprietary or copyrighted maps (e.g. Google Maps, Mapquest) and the terms of use for your map base layer must be compatible with our CC-BY 4.0 license. 

Additional Editor Comments (if provided):

The authors need to revised the following points:

Methods: Details of ethics committee are missing.

Results:

Table 1: The authors have define anemia Anemic (Hb>=11 g/dl) which is not correct. Not sure if the data is appropriate for anemic or nonanemic women. The proportions should indicate prevalence of anemia in the given subgroups. Once anemic prevalence is presented , there is no need for non-anemia prevalence. This is overlapping data.

Table 2: The data should be shown trimester wise for better granularity.

Figure 3: The proportions of mild, moderate and severe anemics do not add to the total anemia prevalence. E.g. North central - anemia in 69.4%. However, mild and moderate anemic fractions add to 97.3%.

Table 3: For logistic regression, the reference should not be extreme in the range. Rather a midpoint should be used. E,g, for wealth quantile, it is better to use middle as reference than poorest.

Discussion: The authors needs to add details of the food groups which were not adequately represented that might be associated with increased risk of anemia.

Conclusion: The conclusion is presently very generic. The author need to specify the exact factors related to dietary diversity and demographic characteristics that are significantly associated with increased risk of anemia.

Reviewers' comments:

Reviewer's Responses to Questions

**Comments to the Author**

1. Does this manuscript meet PLOS Global Public Health’s publication criteria ? Is the manuscript technically sound, and do the data support the conclusions? The manuscript must describe methodologically and ethically rigorous research with conclusions that are appropriately drawn based on the data presented.

Reviewer #1: Yes

2. Has the statistical analysis been performed appropriately and rigorously?

Reviewer #1: Yes

3. Have the authors made all data underlying the findings in their manuscript fully available (please refer to the Data Availability Statement at the start of the manuscript PDF file)?

Reviewer #1: Yes

4. Is the manuscript presented in an intelligible fashion and written in standard English?

Reviewer #1: Yes

5. Review Comments to the Author

Reviewer #1: Introduction

1. Please talk in brief about dietary diversity in Nigeria and how it is defined

Methods:

1. Use consistent terminology: reproductive-age women OR women aged 15-49.

2. 15-49 unit is missing?

3. Line 110 to 114 please move to study population section.

4. Line 118 says providing consent… it should be providing written informed consent

5. Section on study area and its characteristics is missing

6. Who and how data was collected in DHS is not mentioned. Hard copy or e-data collection? What about data quality check? How data managed?

7. Pregnant women height and weight data would have added value. If data is available, please include

8. Morbidity data?

9. Which ethical committee approval were sought?

10. Cite reference for wealth quintile (poorest, poorer, middle, richer and richest)

Results:

1. How many women available in the said population? How many of it approached? Consented? Screened? Enrolled? Screen failure? Withdrawal? Please mention this in very first para of results section.

2. Table 1, age groups- 15-20 and 20-29 are overlapping, please rectify

3. How women below 18 years ages were consented for the study?

4. Table 2, column heading % is mentioned, however CI missing

5. Figure S1 and 4 there is no consistency in which food groups reported/mentioned. Please rectify

6. Figure 4, why data for only 1502 presented?

7. Figure 4 showed data for overall Nigeria as well, but figure S1 showed data for regions only

8. Figure S1, p value given for each food group, however, in figure 4, p value overall given?

6. PLOS authors have the option to publish the peer review history of their article (what does this mean? ). If published, this will include your full peer review and any attached files.

**Do you want your identity to be public for this peer review?** For information about this choice, including consent withdrawal, please see our Privacy Policy .

Reviewer #1: **Yes: ** Dr Dhiraj Agarwal

---

## [Decision Letter · Decision Letter 1]

12 Mar 2025

PGPH-D-24-02093R1

Dietary diversity insufficiently explains differences in prevalence of anaemia in pregnancy across regions in Nigeria: a secondary analysis of Demographic and Health Survey 2018

Dear Dr. Ochuwa Adiketu Babah,

Thank you for submitting your manuscript to PLOS Global Public Health. After careful consideration, we feel that it has merit but does not fully meet PLOS Global Public Health’s publication criteria as it currently stands. Therefore, we invite you to submit a revised version of the manuscript that addresses the points raised during the review process.

We look forward to receiving your revised manuscript.

Kind regards,

Aditi Apte, MD PhD

Academic Editor

Journal Requirements:

Additional Editor Comments (if provided):

The authors have answered majority of the comments. Following minor corrections are needed:

1. Table 1 and table 2 : Please specify the relevance of the p values in the footnote. Are these p values for multiple group comparison.

2. Table 3: For wealth quantiles, poorest is used as a reference for the logistic regression. Poorest is an extreme category. Please use middle as the reference.

Reviewers' comments:

Reviewer's Responses to Questions

**Comments to the Author**

1. If the authors have adequately addressed your comments raised in a previous round of review and you feel that this manuscript is now acceptable for publication, you may indicate that here to bypass the “Comments to the Author” section, enter your conflict of interest statement in the “Confidential to Editor” section, and submit your "Accept" recommendation.

Reviewer #1: All comments have been addressed

2. Does this manuscript meet PLOS Global Public Health’s publication criteria ? Is the manuscript technically sound, and do the data support the conclusions? The manuscript must describe methodologically and ethically rigorous research with conclusions that are appropriately drawn based on the data presented.

Reviewer #1: Yes

3. Has the statistical analysis been performed appropriately and rigorously?

Reviewer #1: Yes

4. Have the authors made all data underlying the findings in their manuscript fully available (please refer to the Data Availability Statement at the start of the manuscript PDF file)?

Reviewer #1: Yes

5. Is the manuscript presented in an intelligible fashion and written in standard English?

Reviewer #1: Yes

6. Review Comments to the Author

Reviewer #1: None

7. PLOS authors have the option to publish the peer review history of their article (what does this mean? ). If published, this will include your full peer review and any attached files.

**Do you want your identity to be public for this peer review?** For information about this choice, including consent withdrawal, please see our Privacy Policy .

Reviewer #1: **Yes: ** Dhiraj Agarwal

---

## [Editor Report · Decision Letter 2]

1 Apr 2025

Dietary diversity insufficiently explains differences in prevalence of anaemia in pregnancy across regions in Nigeria: a secondary analysis of Demographic and Health Survey 2018

PGPH-D-24-02093R2

Dear Ochuwa Adiketu Babah,

We are pleased to inform you that your manuscript 'Dietary diversity insufficiently explains differences in prevalence of anaemia in pregnancy across regions in Nigeria: a secondary analysis of Demographic and Health Survey 2018' has been provisionally accepted for publication in PLOS Global Public Health.

Best regards,

Aditi Apte, MD PhD

Academic Editor